# Research on the Relation between Slump Flow and Yield Stress of Ultra-High Performance Concrete Mixtures

**DOI:** 10.3390/ma15228104

**Published:** 2022-11-16

**Authors:** Jizhong Liu, Mingzhe An, Yue Wang, Song Han, Ziruo Yu

**Affiliations:** School of Civil Engineering, Beijing Jiaotong University, Haidian District, Beijing 100044, China

**Keywords:** ultra-high performance concrete, slump flow, yield stress, rheological property, prediction model

## Abstract

The relation between slump flow and yield stress of ultra-high performance concrete (UHPC) mixtures was studied with theoretical analysis and experimentation. The relational expression between slump flow and yield stress of UHPC mixtures was built and then verified with a rheological test. The results showed that the prediction model, as a function of cone geometry of dimensionless slump flow and dimensionless yield stress of the UHPC mixtures, was constructed based on Tresca criteria, considering the geometric relation of morphological characterization parameters before and after slump of the UHPC mixtures. The rationality and applicability of the dimensionless prediction model was verified with a rheological test and a slump test of UHPC mixtures with different dosages of polycarboxylate superplasticizer. With increase in polycarboxylate superplasticizer dosage, yield stress of the two series of UHPC mixtures (large/small binding material consumption) gradually decreased, leading to a gradual increase in slump flow. Based on the prediction model of dimensionless slump flow and dimensionless yield stress, the relational expression between slump flow and yield stress of the UHPC mixtures was built. The comparison result showed that the calculated data was consistent with the experimental data, which provided a new method for predicting yield stress of UHPC mixtures with a slump test.

## 1. Introduction

Rheological properties of ultra-high performance concrete (UHPC) mixtures, as an index to evaluate flow and deformation capacity of each mixture [1,2,3], directly affect workability, compactness, uniformity and various properties after hardening [4]. The rheological properties of a mixture are characterized by yield stress and plastic viscosity.

The rheological properties of UHPC mixtures are affected by binding material, aggregate, fiber, chemical admixture, mixing method and ambient temperature [5]. Appropriate rheological parameters are necessary for uniform distribution of steel fibers in UHPC. Larger rheological parameters can increase difficulty of uniform distribution of steel fibers during the mixing process, while smaller rheological parameters can lead to significant segregation and settlement of fibers during the pouring process. Meng et al. [6] found that steel fiber was most evenly distributed in the mixture and UHPC had the best bending resistance when the plastic viscosity of UHPC slurry was 53 Pa·s. The rheological parameters of UHPC also have great influence on the aggregate settlement and bubble distribution of the mixture. Smaller rheological parameters cause the aggregate to present a distribution pattern that is ‘upper sparse and bottom dense’ in the mixture, thus affecting mechanical properties and durability of hardened concrete, while smaller plastic viscosity can lead to more bubbles in the mixture, resulting in more pores in the matrix and thus reducing mechanical properties of UHPC [7].

At present, a concrete rheometer is used to obtain the rheological parameters of a given mixture. However, the rheometer test for concrete exhibits many problems, such as complicated operation, high cost of equipment and easy damage. Therefore, many scholars have explored different methods to predict and obtain the rheological parameters of a mixture. Nguyen et al. [8] deduced the mapping function between concrete proportion parameters and rheological parameters of a mixture by using the least square support vector machine algorithm, and established the prediction model of rheological parameters of a mixture based on the mapping function. Mahmoodzadeh et al. [9] built a rheological model using the cell method, considering the material composition of the mixture, and the yield stress and plastic viscosity predicted by the model were in good agreement with the experimental data. By calculating the effective volume fraction of the concrete aggregate, Lee et al. [10] created a rheological model with that volume fraction and wrapped slurry thickness as variable parameters, based on which the yield stress of high-flow concrete was predicted. Ghanbari et al. [11] established a micromechanical constitutive model according to the measured plastic viscosity of self-compacted steel fiber-reinforced concrete mixtures. Cao et al. [12] used the least square method to establish the relation between flowability and rheological properties of a mixture, then obtained the negative exponential relation between slump flow and yield stress and that between flow rate and plastic viscosity, with correlation coefficients of 0.81 and 0.73, respectively. Most flow parameters or prediction models were obtained based on numerical simulations or tests [13,14,15], but there are few studies on the relation between mixture flowability and rheological parameters through theoretical analysis.

UHPC has characteristics such as low water–binder ratio, high amount of binding material, fiber incorporation and lack of coarse aggregate [16,17]; therefore, the rheological properties of UHPC mixtures are quite different from that of ordinary concrete, and the rheological model and prediction formula for ordinary concrete are therefore not suitable for UHPC [18]. At present, there are two main test methods for UHPC mixtures performance. One is the slump test [19], which has a simple operation method, but its test parameters are single, and it is difficult to reflect internal interaction in the mixture. The other method is the rheological performance test [20], which can better characterize the macroscopic working performance of a mixture. This method has little dependence on manual operation and high accuracy of test results, and it can describe movement and deformation of the mixture. However, the rheometer has a high cost and is easy to damage, so it is not suitable for applications on an engineering site. The two methods of mixture performance test are very different, as are their conditions of use. In order to make the two test results consistent and reliable, it is necessary to establish a relation between the flow performance and rheological properties of the UHPC mixtures so as to achieve the purpose of obtaining rheological parameters via slump test and so that workability of UHPC mixtures can be controlled scientifically [21,22]. However, there is a lack of research regarding the prediction model of slump flow and yield stress of UHPC mixtures, so it was necessary to set up a relational model between the measured data in the field test and the rheological properties of the UHPC mixtures. In this paper, based on Tresca criteria, the relational expression between slump flow and yield stress of UHPC mixtures was built, and the rationality and applicability of the relational expression were verified by the rheological test and the slump test.

## 2. Materials and Methods

### 2.1. Raw Materials

The cement used in this study was P·O 42.5 ordinary Portland cement with a fineness of 3400 cm^3^·g^−1^. Silica fume with SiO_2_ content of 90.4% was used; its specific surface area was 14,310 m^2^·kg^−1^. Three different types of quartz sand (coarse, medium, fine) were used, with grain size ranges of 0.315–0.63, 0.63–1.25 and 1.25–2.50 mm; fineness modulu of 1.7, 2.6 and 3.4; and blend proportions of 1:4:2, respectively. Aggregate size distribution is shown in Figure 1. In addition, polycarboxylate superplasticizer with a water reduction rate of 38% and a solid content of 40% was used. Copper-plated, straight steel fiber with tensile strength of 2800 MPa was adopted, as well as the length and diameter of steel fiber was 13 mm and 0.22 mm, respectively.

### 2.2. Mix Proportion

In order to verify the rationality of the model established below and its applicability for UHPC mixtures having different rheological properties, two series of UHPC were designed: series A (less binding material system, 866 kg·m^−3^) and series B (more binding material system, 1326 kg·m^−3^). The mix proportions used in the research were based on the published papers in our research group [23,24]. The water–binder ratio of UHPC in series A was 0.2, and the dosages of polycarboxylate superplasticizer were 1.8%, 2.1%, 2.4% and 2.7% of the binding material, respectively. The water–binder ratio of UHPC in series B was 0.18; the dosages of polycarboxylate superplasticizer were 1.0%, 1.1%, 1.2% and 1.3% of the binding material, respectively. The volume dosage of steel fiber was 2% and the mixing ratio of cement, quartz sand, silica fume, steel fiber, water used in this experiment was 1:1.77:0.23:0.23:0.25 in series A and 1:0.75:0.26:0.15:0.23 in series B. The mix proportions of UHPC in this research are shown in Table 1. The 28 d compressive strength of the UHPC (curing with a controlled temperature of T = 20 ± 2 °C and relative humidity > 95%) is shown in Table 2.

### 2.3. Experimental Method

a. The method of the slump test was performed following the standard for test methods of performance on ordinary fresh concrete (GB/T50080). A slump cone with a top diameter of 100 mm, a bottom diameter of 200 mm, a height of 300 mm and a steel plate with 1500 × 1500 mm was used for the slump test.

b. The MCMR portable concrete rheometer was used to determinate the rheological properties of the mixture; its blade radius (*R_i_)* was 125 mm, the blade height was 125 mm and the container radius (*R*_0_) was 140 mm. First, the UHPC mixtures was mixed with the mixing device; then the UHPC mixtures was quickly poured into the rheometer, after which rheological properties could be tested. During the test, rotation speed was increased from 0 rps to a relatively high value and kept at this speed for a period of time in order to break the thixotropic structure. Then, speed was reduced in stages, with each stage held long enough so that torque was balanced before the fall to the next stage. The concentric cylinder structure of the rheometer is shown in Figure 2a, and the test system of the rheological parameter is shown in Figure 2b.

Yield stress and plastic viscosity of the UHPC mixtures were calculated as shown in Equation (1), using the Bingham model and the Reiner–Riwlin Equation [25].
(1)T=G+HN
where T is torque (N⋅m), N is rotation speed (rad/s) and G , H are the intercept and slope of the fitting line of T−N, related to yield stress and plastic viscosity, respectively. In addition, $\tau_{0}$ and η are shown in Equations (2) and (3), respectively.
(2)$$\tau_{0}=\frac{G}{4\pi h}\left(\frac{1}{R_{i}^{2}}-\frac{1}{R_{0}^{2}}\right)\frac{1}{\ln(\frac{R_{0}}{R_{i}})}$$
(3)$$\eta =\frac{H}{8\pi^{2}h}\left(\frac{1}{R_{i}^{2}}-\frac{1}{R_{0}^{2}}\right)$$
where h is height of the test sample, $\tau_{0}$ is yield stress (Pa) and η is plastic viscosity (Pa⋅s).

## 3. Relation Model between Slump Flow and Yield Stress of UHPC

The stress state of a micro-unit on the surface of the mixture during the slump process from state 1 to state 2 (the concrete slump process was complete, and the mixture was in a static state) is shown in Figure 3. It was assumed that the micro-unit (the small black circles in Figure 3) was spherical; the micro-unit was treated as a rigid body.

Friction force (f) suffered by the micro-unit at any time when the mixture slumped from state 1 (friction was $f_{1}$) to state 2 (friction was $f_{2}$) is shown in Equation (4):(4)$$f=\left(\tau_{0}+\eta \frac{d\gamma }{dt}\right)\pi r^{2}$$
where γ is instantaneous strain of the micro-unit and r is the radius of the micro-unit (m).

It was assumed that the micro-unit move is uniformly accelerated during slump (from state 1 to state 2), and acceleration value is a. Motion is shown in Equation (5):(5)$$G\sin\alpha -f=\frac{4\pi r^{3}\rho a}{3}$$
where G is gravity of the micro-unit (N); α is the included angle between the velocity tangent line and the horizontal plane during micro-unit movement; and ρ is density of the mixture (kg·m^−3^).

The included angle between the tangent line at the position of the micro-unit and the horizontal plane after the mixture stops flowing is β. The force balance equation can be expressed as Equation (6).
(6)$$\frac{4\pi r^{3}\rho g\sin\beta }{3}=\tau_{0}\pi r^{2}$$

Equation (6) can be reduced to Equation (7).
(7)$$\tau_{0}=\frac{4r\rho g\sin\beta }{3}$$

Plastic viscosity (η) could be obtained via combination of Equations (4), (5) and (7), as described by Equation (8).
(8)$$\eta =\frac{\frac{4}{3}r\rho g\left(\sin\alpha -\sin\beta -\frac{a}{g}\right)}{\frac{d\gamma }{dt}}$$

According to Equation (8), plastic viscosity (η) directly affected motion acceleration (a) of the micro-unit; a was closely related to flow rate and flow time of the micro-unit. Therefore, plastic viscosity (η) of the mixture was closely related to flow rate and flow time.

Existing studies had shown that slump flow of mixtures is related to yield stress [26]. In this paper, the relational model between slump flow and yield stress of the UHPC mixtures was deduced based on Tresca yield criteria. The schematic diagram of the mixture before slump (shaded part) is shown in Figure 4.

Pressure ($p_{z}$) exerted on any horizontal layer at a distance (z) in the slump cone is given by Equation (9):(9)$$p_{z}=(\frac{\rho g}{3})\left[(h_{t}+z)-\left(\frac{r_{t}^{2}}{r_{z}^{2}}\right)h_{t}\right]$$
where ρ is density of mixture, g is acceleration of gravity, $h_{t}$ is height of the small cone, $r_{t}$ is the top radius of the slump cone and $r_{z}$ is the radius of the cone at z.

Meanwhile, $h_{t}$, $\left(h_{t}+z\right)$ and H are given by Equations (10)–(12):(10)$$h_{t}=\frac{r_{t}}{\tan\theta }$$
(11)$$(h_{t}+z)=\frac{r_{z}}{\tan\theta }$$
(12)$$H=\frac{r_{H}-r_{t}}{\tan\theta }$$
where H is height of the slump cone and $r_{H}$ is the bottom radius of the slump cone.

Through substitution of Equations (10) and (11) into Equation (9), $p_{z}$ can be expressed as Equation (13).
(13)$$p_{z}=\left(\frac{\rho g}{3\tan\theta }\right)\left(\frac{r_{z}^{3}-r_{t}^{3}}{r_{z}^{2}}\right)$$

Based on Tresca criteria [27], the maximum shear stress ($\tau_{z}$) acting on a plane is one half of the applied pressure ($p_{z}$); $\tau_{z}$ can be expressed as Equation (14).
(14)$$\tau_{z}=\frac{p_{z}}{2}=\left(\frac{\rho g}{6\tan\theta }\right)\left(\frac{r_{z}^{3}-r_{t}^{3}}{r_{z}^{2}}\right)$$

In order to conduct dimensionless process of shear stress, ${\tau_{z}}^{\prime }$ can be obtained by $\tau_{z}$ over ρgH, as expressed in Equation (15).
(15)$$\tau_{z}{}^{\prime }=\frac{\tau_{z}}{\rho gH}=\left(\frac{1}{6H\tan\theta }\right)\left(\frac{r_{z}^{3}-r_{t}^{3}}{r_{z}^{2}}\right)$$

With substitution of Equation (12) into Equation (15), ${\tau_{z}}^{\prime }$ is given in Equation (16):(16)$$\tau_{z}{}^{\prime }=\frac{1}{6\left(r_{H}-r_{t}\right)}\left(\frac{r_{z}^{3}-r_{t}^{3}}{r_{z}^{2}}\right)$$

Assuming that the mixture is incompressible and the volume of the micro-unit before slump deformation (state A) is the same as that after slump deformation (state B), as shown in Figure 5.

The volume of the micro-unit did not change during slump deformation, so the corresponding relationship can be expressed as Equation (17):(17)$$\pi d_{r_{z}}^{2}z=\pi d_{r_{z}{}_{1}}^{2}z_{1}$$
where z and $z_{1}$ are height of the micro-unit before and after slump deformation, respectively. In addition, $dr_{z}$ and $dr_{z1}$ are the corresponding radiuses, respectively. Equation (18) can be obtained according to Equation (17).
(18)$$dr_{z_{1}}={\left(\frac{z}{z_{1}}\right)}^{1/2}dr_{z}$$

Slump flow of the mixture is shown in Figure 6. The radius of the mixture changed from $r_{H}$ to $r_{F}$ after slump deformation, and $r_{F}$ can be expressed as Equation (19):
(19)$$r_{F}=r_{H}+r_{0}+r_{1}$$
where $r_{0}$ is the distance between where shear stress equals yield stress and the bottom edge of the slump cone. In addition, $r_{1}$ is the distance between the outer edge of the region where shear stress is less than yield stress and the edge of $r_{0}$. $r_{1}$ is generated by driving force, and the flow distance of the mixture under the driving force is short, therefore, $r_{1}$ ≪ $r_{0}$.

Next, $r_{0}$ can be expressed as Equation (20).
(20)$$r_{0}=\int_{r_{H}}^{r_{F}-r_{1}}dr_{z_{1}}$$

Through substitution of Equation (18) into Equation (20), $r_{0}$ can be expressed as Equation (21).
(21)$$r_{0}={\int_{r_{H}}^{r_{{}_{F}}-r_{1}}\left(\frac{z}{z_{1}}\right)}^{1/2}dr_{z}$$

During the process of slump, the cross-sectional area of the micro-unit increases, but the shear force on the cross section is the same, and its relation can be expressed as Equation (22).
(22)$$\tau_{z}\left(\pi d_{r_{z}}^{2}\right)=\tau_{z_{1}}\left(\pi d_{r_{z_{1}}}^{2}\right)$$

When shear stress on the section was equal to yield stress, the relation could be expressed as Equation (23).
(23)$$\tau_{z_{1}}\left(\pi d_{r_{z_{1}}}^{2}\right)=\tau_{0}\left(\pi d_{r_{z_{1}}}^{2}\right)$$

Equation (24) is obtained from Equations (22) and (23).
(24)$$d_{r_{z_{1}}}={\left(\frac{\tau_{z}}{\tau_{0}}\right)}^{1/2}dr_{z}$$

Through combination of Equations (18) and (24), Equation (25) is given.
(25)$$\frac{z}{z_{1}}=\frac{\tau_{z}}{\tau_{0}}$$

Through substitution of Equation (25) into Equation (21) and with reference to the literature [28], Equation (26) is given.
(26)$$r_{0}={\int_{r_{H}}^{r_{{}_{F}}-r_{1}}\left(\frac{\tau_{z}}{\tau_{0}}\right)}^{1/2}dr_{z}={\int_{r_{H}}^{r_{{}_{F}}-r_{1}}\left(\frac{{\tau_{z}}^{\prime }}{{\tau_{0}}^{\prime }}\right)}^{1/2}dr_{z}$$

Through substitution of Equation (16) into Equation (26), Equation (27) is given.
(27)$$r_{0}={\int_{r_{H}}^{r_{F}-r_{1}}\left[\frac{\frac{1}{\left(r_{H}-r_{t}\right)}\left(\frac{r_{z}^{3}-r_{t}^{3}}{r_{z}^{2}}\right)}{6{\tau_{0}}^{\prime }}\right]}^{1/2}dr_{z}$$

According to Equation (19), ${r_{F}}^{\prime }$ can be expressed as Equation (28).
(28)$${r_{F}}^{\prime }=\frac{r_{F}}{r_{H}}=1+\frac{r_{0}}{r_{H}}+\frac{r_{1}}{r_{H}}$$

According to Equation (16), dimensionless yield stress (${\tau_{0}}^{\prime }$) can be expressed as Equation (29).
(29)$${\tau_{0}}^{\prime }=\frac{1}{6\left(r_{H}-r_{t}\right)}\left(\frac{r_{0}^{3}-r_{t}^{3}}{r_{0}^{2}}\right)$$

Through combination of Equation (19) and Equations (27)–(29), ${\tau_{0}}^{\prime }$ can be simplified as Equation (30).
(30)$${\tau_{0}}^{\prime }=\frac{1}{20{r_{F}}^{\prime }{}^{2}-20{r_{F}}^{\prime }}\left({r_{F}}^{\prime }\;>\;1\right)$$

According to Equation (30), dimensionless yield stress (${\tau_{0}}^{\prime }$) can be characterized by dimensionless slump flow (${r_{F}}^{\prime }$) so as to achieve the purpose of obtaining yield stress of the UHPC mixture via slump test.

## 4. Experimental Results and Model Validation

### 4.1. Experimental Results

The torque–rotation speed curves of two series of UHPC mixtures (series A and series B) are shown in Figure 7. The curve of the UHPC mixtures under different dosages of polycarboxylate superplasticizer presented a linear relation, which interpreted the UHPC mixtures belong to Bingham fluid [29]. Although the dosage of polycarboxylate superplasticizer of series A was greater than that of series B, the plastic viscosity of series A was greater than that of series B (the torque–rotation curve slope of series A was higher than that of series B) due to series A having a lower binder–sand ratio as compared with series B. Plastic viscosity was 2847, 2719, 1965 and 1628 Pa·s when the dosage of polycarboxylate superplasticizer for the series A mixture was 1.8%, 2.1%, 2.4% and 2.7%, respectively. The polycarboxylate superplasticizer dosage of the series B mixture was 1.0%, 1.1%, 1.2% and 1.3%, respectively, meanwhile the plastic viscosity was 1220, 894, 589 and 255 Pa·s, respectively.

Figure 8 shows the effect of polycarboxylate superplasticizer dosage on yield stress of two series of UHPC mixtures. It can be seen from Figure 8 that in series A and B, yield stress of the UHPC mixtures decreased successively with increase in dosage of polycarboxylate superplasticizer.

Two series of UHPC had different water–binder ratios, binding material dosages and sand mixing ratios, so the degree of influence of polycarboxylate superplasticizer dosage on yield stress of a given UHPC mixtures was different. In series A, yield stress of the UHPC mixtures was 437, 399, 48 and 21 Pa when the dosage of polycarboxylate superplasticizer was 1.8%, 2.1%, 2.4% and 2.7%, respectively. Yield stress of the UHPC mixtures decreased by 8.7% when the dosage of polycarboxylate superplasticizer increased from 1.8% to 2.1%. Yield stress decreased significantly, by 87.9%, when the dosage of polycarboxylate superplasticizer increased from 2.1% to 2.4%. Yield stress of the UHPC mixtures decreased by 56.3% when the dosage of polycarboxylate superplasticizer increased from 2.4% to 2.7%.

In series B, yield stress of the UHPC mixtures was 817, 181, 147 and 40 Pa when the dosage of polycarboxylate superplasticizer was 1.0%, 1.1%, 1.2% and 1.3%, respectively. Yield stress of the UHPC mixtures decreased significantly by 77.8% when the dosage of polycarboxylate superplasticizer increased from 1.0% to 1.1%. Subsequently, yield stress of the UHPC mixtures decreased slowly with increase in dosage of polycarboxylate superplasticizer. Yield stress decreased by 18.8% when the dosage of polycarboxylate superplasticizer increased from 1.1% to 1.2%. Yield stress decreased by 56.3% when the dosage of polycarboxylate superplasticizer increased from 1.2% to 1.3%. Superplasticizer is adsorbed on the surface of cementitious particles so that the suspended particles are far enough away to prevent the particles from being attracted by van der Waals forces. At the same time, superplasticizer made the cementitious particles having a negative charge. When the negatively charged particles repeled each other, the water originally in the condensed particles is released, resulting in an increase of the thickness of the water film on the particles surface, as well an increase of the distance between particles, and also a drop-off of the van der Waals force between particles. Therefore, yield stress of a UHPC mixtures decreases when the dosage of superplasticizers increases [30].

The experiment results for slump flow and yield stress of two series of UHPC mixtures and their relation curves are shown in Figure 9. The states of the UHPC mixtures of A2 and B2 are shown in Figure 10. It can be seen in Figure 9 that the increase in yield stress of the UHPC mixtures led to a gradual decrease of slump flow, and there was no linear relation between yield stress and slump flow, as opposed to what is basically a linear relation between yield stress and slump flow of an ordinary concrete mixture [31,32]. The binding material dosage and water–binder ratio of the two series of UHPC mixtures were different, so the decreasing amplitude of slump flow was different from the increase in yield stress. The slump flow was 425, 395, 331 and 313 mm when the yield stress of series A was 21, 48, 399 and 437 Pa, respectively, and slump flow was 651, 445, 380 and 250 mm when the yield stress of series B was 40, 147, 181 and 817 Pa, respectively.

### 4.2. Model Validation

In order to verify the reasonableness of the dimensionless model, Equations (15) and (28) were used, respectively, and dimensionless results ${r_{F}}^{\prime }$ and ${\tau_{0}}^{\prime }$ were obtained. At the same time, in order to prove the applicability of the dimensionless model to UHPC mixtures and to enrich the comparative data, the flow radius ($r_{F}$) and yield stress ($\tau_{0}$) in references [12,26,33,34,35,36] were processed. The dimensionless data are shown in Figure 11, and it can be seen in the figure that ${\tau_{0}}^{\prime }$ is coincident with the dimensionless yield stress model curve. However, the experiments in the references used different types of chemical admixture, indicating that the prediction model is not dependent on type of chemical admixture and that the dimensionless yield stress model can be used to predict yield stress of UHPC mixtures.

The model above is a dimensionless yield stress model. In order to obtain the absolute value of yield stress of a UHPC mixtures, Equations (15) and (28) were substituted into Equation (30) to obtain the relational expression between slump flow and yield stress of the UHPC mixtures, as shown in Equation (31).
(31)$$\tau_{0}=\frac{r_{H}^{2}\rho gH}{20\left(r_{F}^{2}-r_{F}r_{H}\right)}$$

In order to prove rationality and applicability of the relational expression between slump flow and yield stress of the UHPC mixtures, the calculated yield stress data of UHPC mixtures A1, A2, B2, B3 and B4 were compared with the experimental data; the states of the UHPC mixtures of A3 and A4 were too diluted due to the large dosage of polycarboxylate superplasticizer. However, the state of UHPC mixtures of B1 was too dry due to the small dosage of polycarboxylate superplasticizer. Therefore, these three groups were not representative. The comparison result is shown in Figure 12. Bland–Altman consistency analysis was carried out by using Medcalc software to analyze difference between calculated data and experimental data. The statistical results are shown in Figure 13. It can be seen from Figure 13 that the differences between calculated data and experimental data are all within 95% confidence interval, indicating that the calculated yield stress values obtained by the relational expression are consistent with those of the experimental data, and the relational expression is credible, indicating that the relational expression between slump flow and yield stress can predict yield stress of UHPC mixtures accurately.

## 5. Conclusions

The relation between slump flow and yield stress of UHPC mixtures was studied via theoretical analysis and experiment, and the following conclusions could be drawn:

(1) The prediction model as a function of cone geometry of dimensionless slump flow and dimensionless yield stress of a UHPC mixtures was constructed based on Tresca criteria and considering the geometric relation of morphological characterization parameters before and after slump of the UHPC mixtures. The rationality and applicability of the dimensionless prediction model were verified with rheological testing and slump testing of the UHPC mixtures with different dosages of binding material, indicating that the dimensionless prediction model can be used for theoretical calculation of yield stress of the UHPC mixtures. The relational expression between slump flow and yield stress of the UHPC mixtures was built.

(2) Yield stress of the two series of UHPC (large/small binding material consumption) decreased gradually with increase in dosage of polycarboxylate superplasticizer. The decrease of yield stress resulted in gradual increase in slump flow, but there was no linear relation between yield stress and slump flow. Different amounts of binding material resulted in different degrees of effect of polycarboxylate superplasticizer on slump flow of the UHPC mixtures. The calculated yield stress was compared with the experimental data and the comparison result showed that the relational expression is suitable for a UHPC mixtures with a low water–binder ratio and a high/low sand–binder ratio.

(3) This research exploratively found a method to predict yield stress of the UHPC mixtures via slump test, which laid part of the theoretical foundation for further, related research. Based on the relational expression, yield stress of a UHPC mixtures can be obtained using height of the slump cone, bottom radius of the slump cone, flow radius of the mixture and density of the mixture. In particular, it is worth noting that steel fiber volume content should be considered when developing further research about the relation between slump flow and yield stress of UHPC mixtures.

## Figures and Tables

**Figure 1 materials-15-08104-f001:**
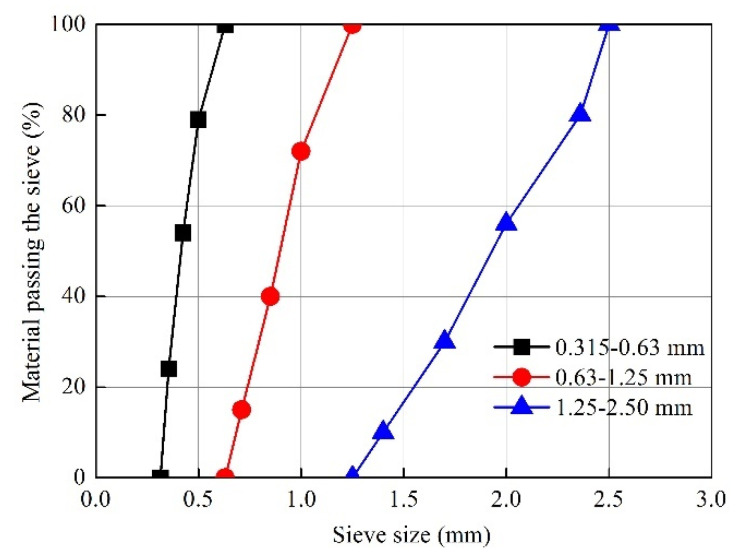
Sand gradations.

**Figure 2 materials-15-08104-f002:**
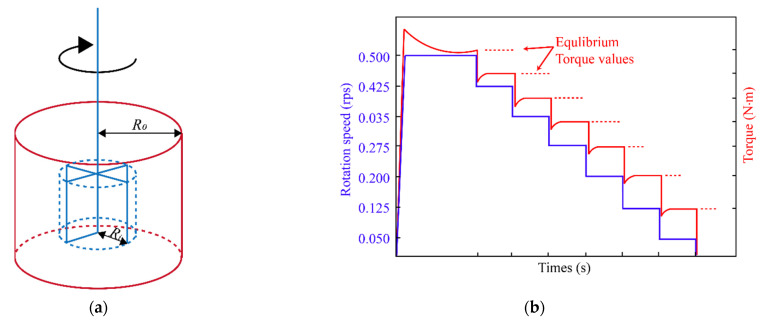
(**a**) Concentric cylinder structure of rheometer. (**b**) Test system of rheological parameter.

**Figure 3 materials-15-08104-f003:**
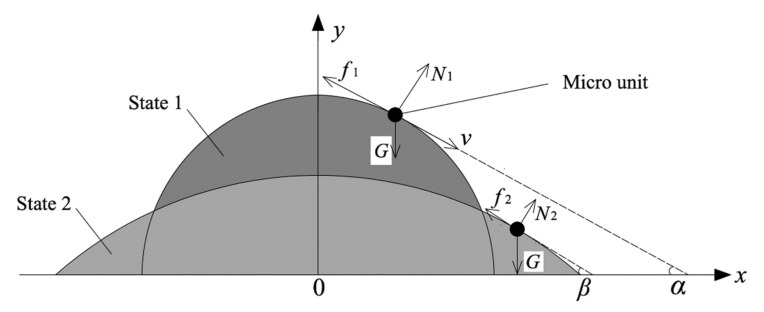
Stress state of micro-unit of mixture in slump process.

**Figure 4 materials-15-08104-f004:**
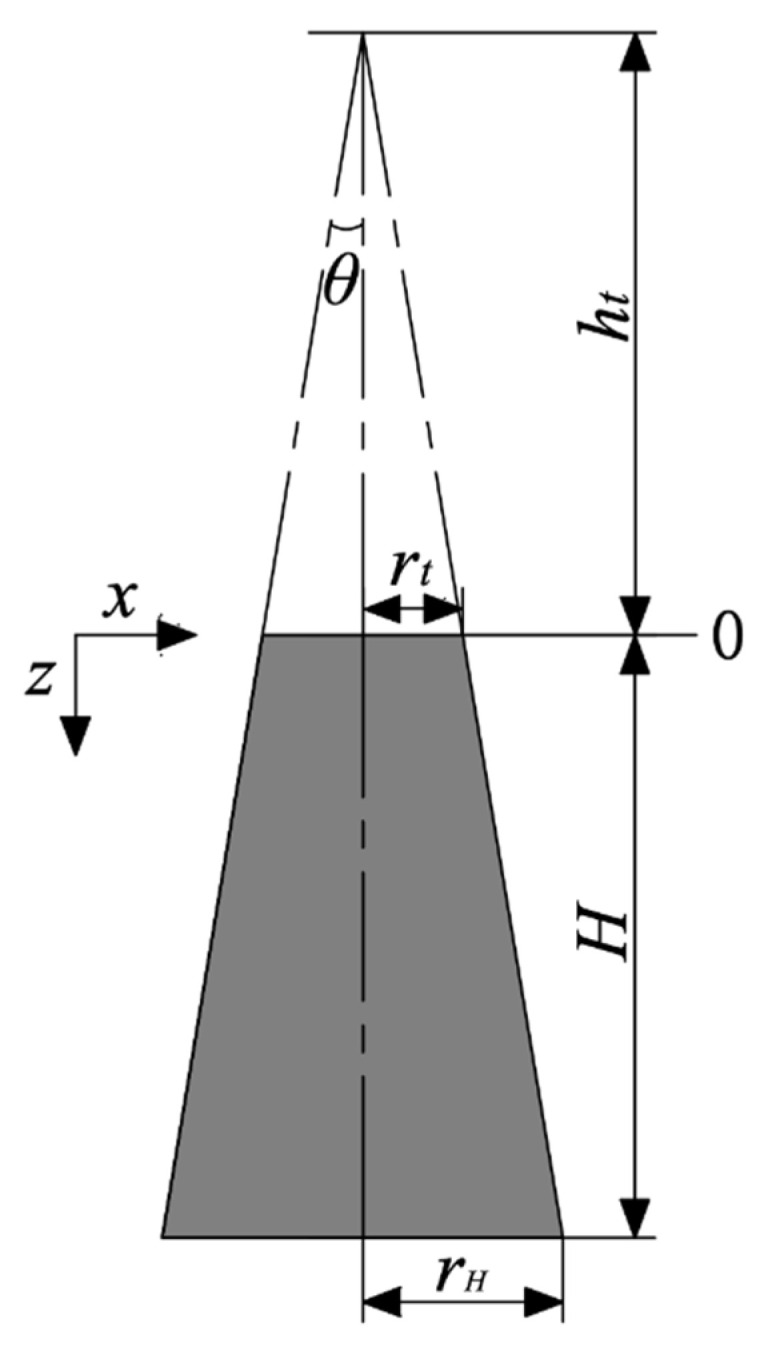
Schematic diagram of mixture before slump.

**Figure 5 materials-15-08104-f005:**
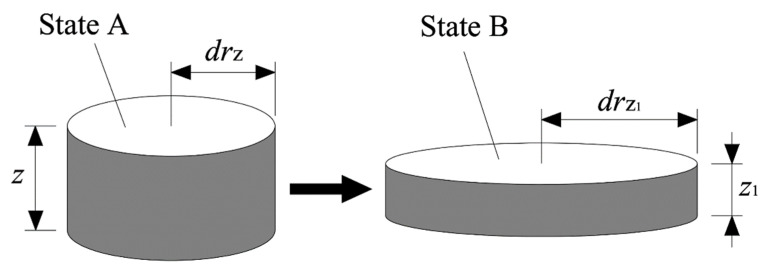
Schematic of micro-unit variation (from state A to state B).

**Figure 6 materials-15-08104-f006:**
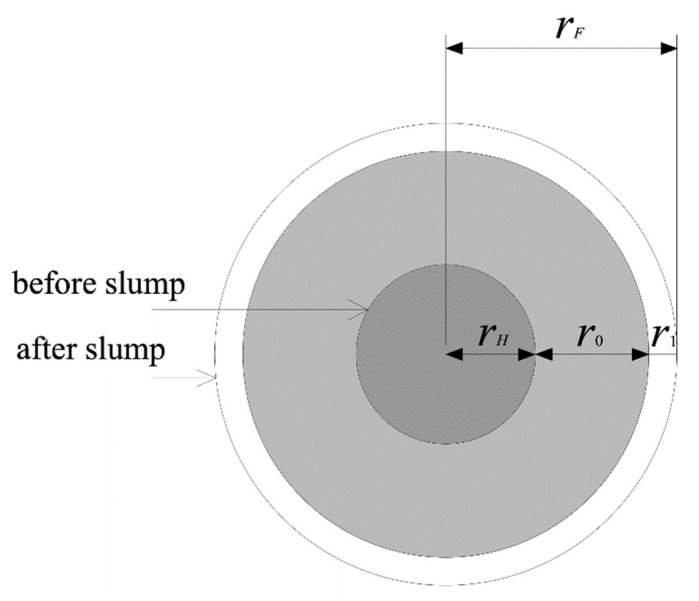
Diagram of mixture slump flow.

**Figure 7 materials-15-08104-f007:**
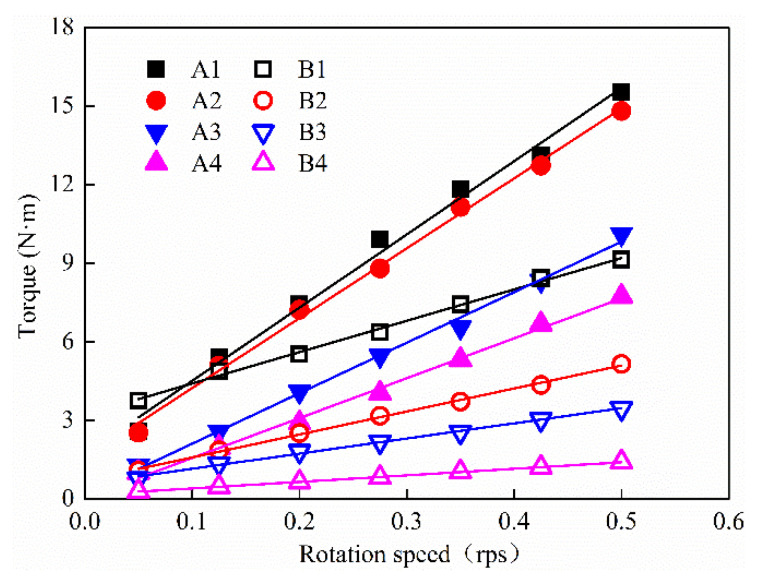
Torque–rotation speed curves of UHPC.

**Figure 8 materials-15-08104-f008:**
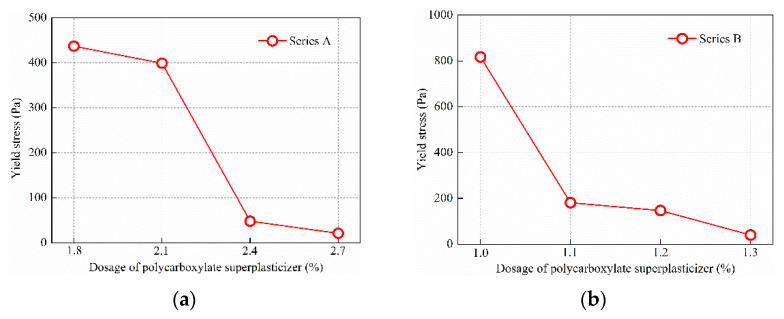
(**a**) Influence of polycarboxylate superplasticizer dosage on yield stress of series A UHPC. (**b**) Influence of polycarboxylate superplasticizer dosage on yield stress of series B UHPC.

**Figure 9 materials-15-08104-f009:**
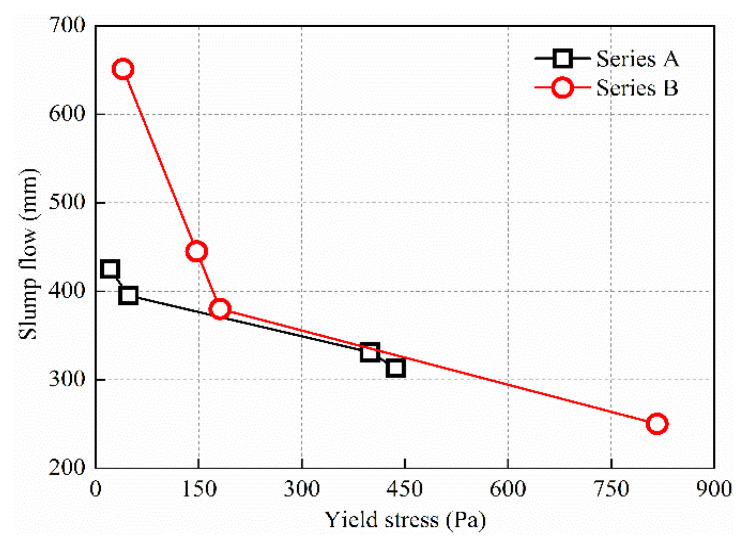
Relation between slump flow and yield stress of UHPC mixture.

**Figure 10 materials-15-08104-f010:**
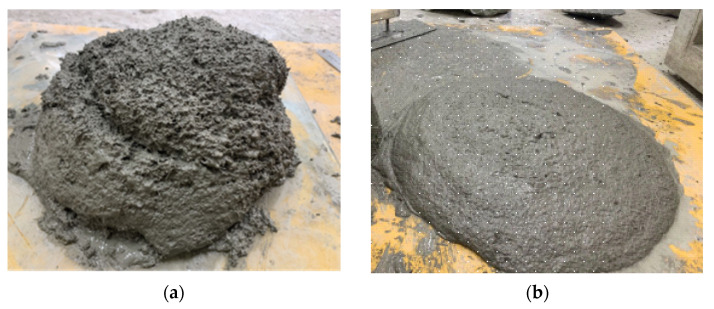
(**a**) UHPC mixture of A2. (**b**) UHPC mixture of B2.

**Figure 11 materials-15-08104-f011:**
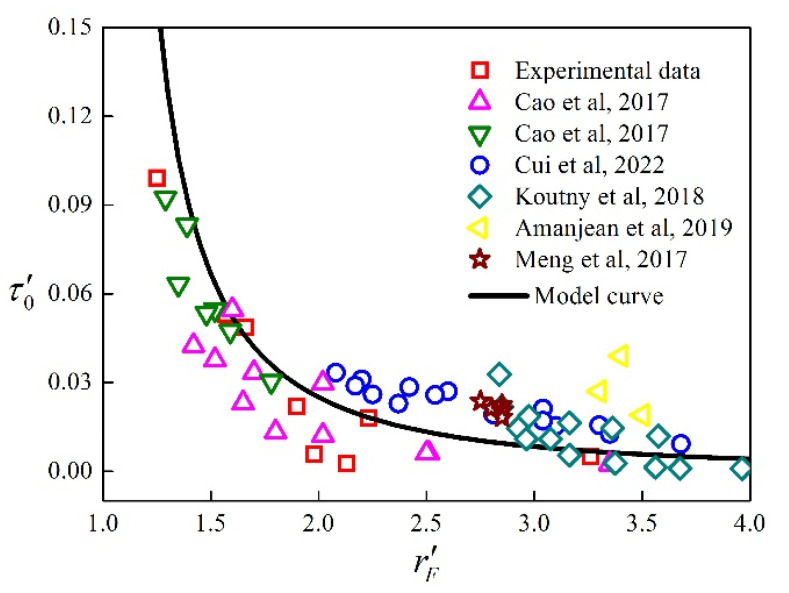
Comparison between dimensionless model curve and experiment data [12,26,33,34,35,36].

**Figure 12 materials-15-08104-f012:**
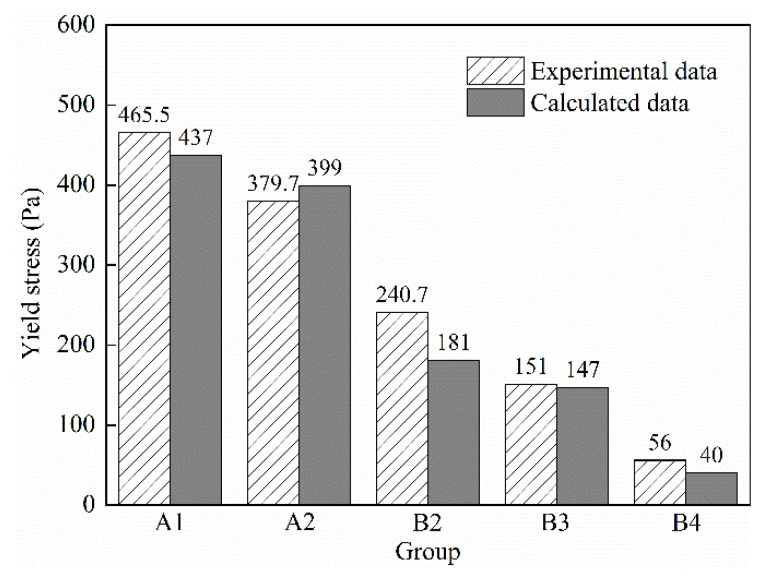
Comparison between calculated data and experimental data of yield stress.

**Figure 13 materials-15-08104-f013:**
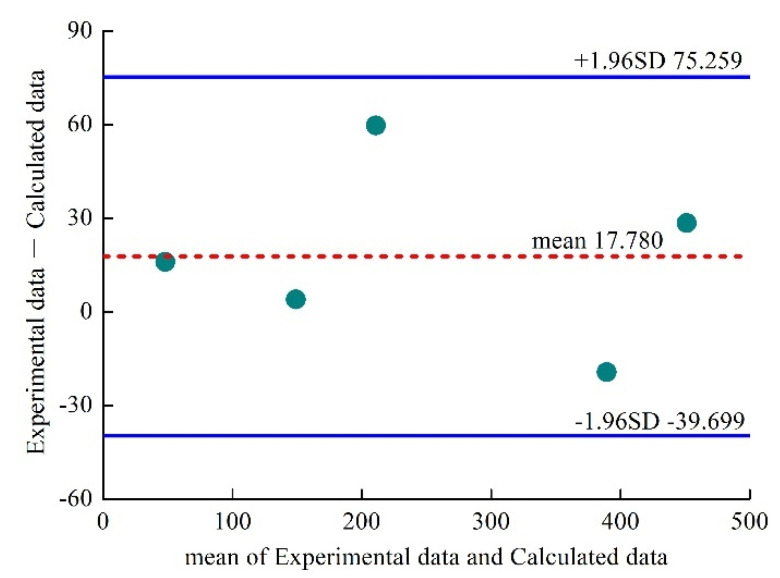
Bland–Altman consistency analysis.

**Table 1 materials-15-08104-t001:** Mix proportions of UHPC (kg·m^−3^).

Type	Cement	Silica Fume	Quartz Sand			Water	Polycarboxylate Superplasticizer	Steel Fiber
			Coarse	Medium	Fine			
A1	706	160	357	714	178	164	15.59	160
A2	706	160	357	714	178	162	18.19	160
A3	706	160	357	714	178	161	20.78	160
A4	706	160	357	714	178	159	23.38	160
B1	1050	276	225	449	113	231	13.26	160
B2	1050	276	225	449	113	230	14.59	160
B3	1050	276	225	449	113	229	15.91	160
B4	1050	276	225	449	113	228	17.24	160

UHPC stands for ultra-high performance concrete. In A1, A2, A3 and A4, A stands for series A and 1, 2, 3 and 4 stand for dosage of polycarboxylate superplasticizer: 1.8%, 2.1%, 2.4% and 2.7% of the binding material, respectively. In B1, B2, B3 and B4, B stands for series B and 1, 2, 3 and 4 stand for dosage of polycarboxylate superplasticizer: 1.0%, 1.1%, 1.2% and 1.3% of the binding material, respectively.

**Table 2 materials-15-08104-t002:** 28 d compressive strength of UHPC (MPa).

Type	A1	A2	A3	A4	B1	B2	B3	B4
Compressive strength	107.3	111.5	112.1	106.4	106.1	109.3	105.2	104.6

## Data Availability

Data is contained within the article.

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
