# Peer review of "Research on the Relation between Slump Flow and Yield Stress of Ultra-High Performance Concrete Mixtures"

_materials, 2022, doi:10.3390/ma15228104_

Round 1

Reviewer 2 Report

The authors present a very good paper related to the "Relation Between Slump Flow and Yield Stress of Ultra-High Performance Concrete Mixture". In general, the paper is well-structured and well-written, the arguments are clearly developed and the conclusions are soundly-based.  This is an interesting paper but some modifications must be carried out in order to improve the global quality of paper, namely:

1. A new subsection should be created in the Introduction section, "Research significance and objectives", in order to explain the main objectives and the novelty associated with this work.

2. Please explain in more detail the experimental method used.

3. The experimental results should be compared with more results present in the literature. The model proposed should be compared with more results present in the literature.

4. A statistical analysis should be presented in the Results section.

Finally, I believe the work is suitable for publication in the Materials journal, after minor revisions.

Reviewer 3 Report

1. Specify/justify in the introduction how the results of this theoretical study can be used in practice. When can the developed prediction model be used in practice?     

2. There is not any proof that the mixtures in Table 1 all are UHPC. Just mix proportions cannot represent the quality of a mixture.

3. How mix proportions were designed? Is the cement dosage in both series within the normal range of UHPC? Please justify with supporting references.

4. The type of superplasticizer and its dosage in the mixture are very important in the Slump Flow and Yield Stress, as specified in the second conclusion. In this research, a specific type of superplasticizer was used. If another type or generation of the superplasticizer is used, then the theory and experimental test results could be different. The authors should specify and discuss whether the prediction model is dependent on the type of chemical admixture or not.  

5. Give a few photos from the results of flow tests.

6. Give a brief introduction to section 5.

Reviewer 4 Report

The Revision of the Manuscript entitled: "Research on the Relation Between Slump Flow and Yield Stress of Ultra-High Performance Concrete Mixture".

The authors presented very interesting scientific work. The structure of the Manuscript is sufficient for such evaluation. The methodology is described in such way that someone who wants to repeat the study, can do that. Results are in common with the literature.

However some issues should be revised before publishing this Manuscript:

- The authors should be carefull with  such statements as "ultra-low". What does it mean? w/b = 0.001? or 0.1 or what value?

-The last paragraph of the introduction should be strenghten with the references to prove the motivation of the study. These sentences / research questions should be the extension of the literature survey. But now the last paragraph is not sufficiently connected with the introduction.

- Presenting the chart of aggregate size distribution would be beneficial.

- The Authors compared in figure 10 only 5 groups not 8. Why? 

The authors need to rework the conclusions section. Now it is presenting only the repeated analyses of the results. It has to answer the questions stated in the introduction about the motivation of the study. 

- In conclusions the authors should point out the  possibilities of further research

Round 2

Reviewer 3 Report

The authors addressed my comments, however still needs two more actions:

1. My comment #2; at least compressive strength of concrete mixtures should be reported. 

2. My comment #3; descriptions and references should be in section 2.2.

Reviewer 4 Report

The article significantly improved from the revision.
